# Predictors of hospital expenses and hospital stay among patients undergoing total laryngectomy: Cost effectiveness analysis

**Ming-Hsien Tsai[1,2,3], Hui-Ching Chuang[1,2], Yu-Tsai Lin[1,2], Hui Lu[1], Fu-Min Fang[2,4], Tai-Lin Huang[2,5], Tai-Jan Chiu[2,5], Shau-Hsuan Li[2,5], Chih-Yen Chien**[1,2,6]*

**1** Department of Otolaryngology, Kaohsiung Chang Gung Memorial Hospital and Chang Gung University College of Medicine, Kaohsiung, Taiwan, **2** Kaohsiung Chang Gung Head and Neck Oncology Group, Cancer Center, Kaohsiung Chang Gung Memorial Hospital, Kaohsiung, Taiwan, **3** College of Pharmacy and Health Care, Tajen University, Pingtung County, Taiwan, **4** Department of Radiation Oncology, Kaohsiung Chang Gung Memorial Hospital and Chang Gung University College of Medicine, Kaohsiung, Taiwan, **5** Department of Hematology and Oncology, Kaohsiung Chang Gung Memorial Hospital and Chang Gung University College of Medicine, Kaohsiung, Taiwan, **6** Institute For Translational Research In Biomedicine, Kaohsiung Chang Gung Memorial Hospital, Kaohsiung, Taiwan

* cychien3965@adm.cgmh.org.tw

**Data Availability Statement:** All relevant data are within the manuscript and its Supporting Information files. And All relevant data are held in

## Abstract

### Objective

To determine the predictive factors of postoperative hospital stay and total hospital medical cost among patients who underwent total laryngectomy.

### Methods

A total of 213 patients who underwent total laryngectomy in a tertiary referral center for tumor ablation were enrolled retrospectively between January 2009 and May 2018. Statistical analyses including Pearson's chi-squared test were used to determine whether there was a significant difference between each selected clinical factors and outcomes. The outcomes of interest including postoperative length of hospital stay and inpatient total medical cost. Logistic regression analyses were performed to reveal the relationship between clinical factors and postoperative length of hospital stay or total inpatient medical cost.

### Results

Preoperative radiotherapy ($p = 0.007$), method of wound closure ($p < 0.001$), postoperative serum albumin level ($p = 0.025$), and postoperative serum hemoglobin level ($p = 0.04$) were significantly associated with postoperative hospital stay in univariate analysis. Postoperative hypoalbuminemia (odds ratio [OR]: 2.477; 95% confidence interval [CI]: 1.189–5.163; $p = 0.015$) and previous radiotherapy history (OR 2.194; 95% CI: 1.228–3.917; $p = 0.008$) are independent predictors of a longer postoperative hospital stay in multiple regression analysis. With respect to total inpatient medical cost, method of wound closure ($p < 0.001$), preoperative serum albumin level ($p = 0.04$), postoperative serum albumin level ($p < 0.001$), and history of liver cirrhosis ($p = 0.037$) were significantly associated with total inpatient medical

Figshare, a public repository. https://doi.org/10.6084/m9.figshare.12568565.v1

**Funding:** The author(s) received no specific funding for this work.

**Competing interests:** The authors have declared that no competing interests exist.

cost in univariate analysis. Postoperative hypoalbuminemia (OR: 6.671; 95% CI: 1.927–23.093; $p = 0.003$) and microvascular free flap reconstruction (OR: 5.011; 95% CI: 1.657–15.156; $p = 0.004$) were independent predictors of a higher total inpatient medical cost in multiple regression analysis.

## Conclusions

Postoperative albumin status is a significant factor in predicting prolonged postoperative hospital stay and higher inpatient medical cost among patients who undergo total laryngectomy. In this cohort, the inpatient medical cost was 48% higher and length of stay after surgery was 35% longer among hypoalbuminemia patients.

## Introduction

Current treatment of locally advanced laryngeal cancer/hypopharyngeal cancer has gradually evolved to the strategy of concurrent chemoradiotherapy (CRT). However, total laryngectomy still plays a role in primary advanced T4 laryngeal/hypopharyngeal cancer, in persistent or recurrent tumors after primary radiation (RT) for salvage purposes, or in a non-functional larynx after previous treatment [1–3]. Postoperative wound complications and prolonged hospital stay are important issues in primary and salvage total laryngectomy (TL) [4, 5]. When postoperative major wound complications occur among patients with head and neck squamous cell carcinoma (HNSCC), it delays adjuvant therapy, prolongs hospital stay, increases medical expense, and induces a higher risk of psychological distress among patients. There are several factors related to wound complications, including previous history of CRT/RT, poor nutrition status, anemia, neck dissection, and tumor stage. Currently, it will be worthwhile to realize the factors about the prolonged hospital length of stay (LOS) and higher total inpatient medical cost among patients undergoing total laryngectomy. Therefore, the aim of this study is to determine the predictive factors of postoperative prolonged hospital LOS and higher inpatient medical cost among patients undergoing total laryngectomy.

## Materials and methods

### Study population

Patients who underwent total laryngectomy with or without microvascular free flap reconstruction were enrolled retrospectively from the institutional cancer database between January 2009 and May 2018 in Kaohsiung Chang Gung Memorial hospital, Taiwan. Free flap transfer for pharyngeal defect reconstruction after total laryngectomy would be performed if pharyngeal defect couldn't primarily close without tension. Patients who received partial laryngectomy or partial laryngopharyngectomy or patient didn't receive total laryngectomy surgical procedure in our hospital would be excluded from our study. Treatment was primarily based on the American National Comprehensive Cancer Network (NCCN) guidelines.

Peri-operative clinical variables of interest were collected, including age; sex; performance status (Eastern Cooperative Oncology Group (ECOG) score); primary tumor location and histology; body mass index (BMI); pre-operative serum hemoglobin; pre- and post-operative serum albumin; post-operative serum hemoglobin; past medical history, including diabetes mellitus and liver cirrhosis; prior interventions such as RT and CRT; and reason for TL (i.e., primary tumor, salvage treatment for persistent or recurrent cancer, or non-functional larynx).

Operative details, including completion of neck dissection(s) and method of pharyngeal closure (primary closure, use of free tissue reconstruction), were also included. Post-operative wound complications, LOS, and inpatient medical cost were also reviewed.

## Variables and outcomes

Patients were retrospectively enrolled according to the following clinical characteristics: gender, age, primary tumor site, post-operative wound condition, length of hospital stay, and total medical cost of this hospitalization. BMI and circulatory laboratory data, including serum hemoglobin and serum albumin, were regularly measured within 1 week before the surgery. Postoperative serum hemoglobin is defined as the serum hemoglobin level collected the morning after the surgery. Postoperative serum albumin is defined as the serum albumin level collected the morning after the surgery. Anemia is defined as serum hemoglobin level $< 13$ g/dL. Hypoalbuminemia is defined as serum albumin level $< 3.5$ g/dL. Patients with liver cirrhosis presenting as Child-Pugh C classification were excluded from analysis. Major postoperative wound infection is defined as a postoperative recipient-site wound condition that necessitated wound debridement in the operating room. Length of hospital stay after surgery is defined as the period between the operation and discharge from the ward in this study.

## Statistical analysis

Statistical analyses were performed using SPSS 20.0 software (SPSS/IBM, Inc., Chicago, IL). The endpoints of this study included major postoperative wound infection, length of hospital stay after surgery, and inpatient total medical cost of this hospitalization. A mean and median approach was applied to select appropriate thresholds for hospital stay and medical cost. Pearson's chi-squared test was used to determine whether there was a significant difference between each selected clinical factor and the outcome we were interested in, such as postoperative major wound infection, postoperative length of hospital stay, and inpatient total medical cost. Logistic regression analyses were performed to reveal the relationship between postoperative length of hospital stay or total inpatient medical cost and clinical factors. The estimated odds ratios (ORs) and 95% confidence intervals (CIs) were calculated for each independent factor. To compare the central tendency of hospital stay of the lower postoperative albumin group versus the higher postoperative albumin group, as well as their medical costs, the Mann–Whitney $U$ test was applied. A 2-tailed $p$-value $< 0.05$ was considered significant. This study was approved by the Medical Ethics and Human Clinical Trial Committees at Chang Gung Memorial Hospital (Ethical Application Reference number:201900875B0). Patients' consent to review their medical records was not required by this hospital's committees because the patient data remained anonymous in this study.

## Results

A total of 213 patients were enrolled in this study. The clinical characteristics of the study patients are summarized in Table 1. The patients' median age was 58 years (range: 35–88). The population included 201 (94.4%) male patients and 12 (5.6%) female patients. The ECOG performance status score in our cohort were all 0–1. The average BMI in this population was 21.86 kg/m$^2$ (range: 14.2–32.83). The median length of hospital stay after surgery was 18 days (range: 7–70). The mean length of stay after surgery was 22.48 days, with a standard deviation of 12.49 days (Fig 1). The average total inpatient medical cost was 378,967 New Taiwan dollars (NTD) (range: 121,275–1,192,203 NTD ≈ 4,016–39,477 U.S. dollars ≈ 3,567–35,065 EUR (1 U.S. dollar is roughly equal to 30.2 NTD; 1 Euro is roughly equal to 34.0 NTD, according to the Bank of Taiwan, as of June 3, 2020).

**Table 1. Clinicopathological characteristics of 213 patients underwent total laryngectomy.**

| Characteristics | | Value | % |
|---|---|---|---|
| Age Median(range), yr | | | |
| Sex | Male | 201 | 94.4 |
| | Female | 12 | 5.6 |
| BMI [a] Average (range), Kg/m$^2$ | | 21.86 (14.2 ~ 32.83) | |
| Median hospital length of stay (range), days | | 26 (10 ~ 107) | |
| Median postoperative hospital length of stay (range), days | | 18 (7 ~ 70) | |
| Average of total inpatient medical cost (NTD [b]) | | 378,967 (121,275 ~ 1,192,203) | |
| Average preoperative serum hemoglobin (range), g/dL | | 12.901 (7.70 ~ 17.80) | |
| Average postoperative serum hemoglobin (range), g/dL | | 11.637 (8.35 ~ 15.40) | |
| Average preoperative serum albumin (range), g/dL | | 4.075 (2.70 ~ 4.95) | |
| Average postoperative serum albumin (range), g/dL | | 3.162 (2.00 ~ 4.20) | |
| Diabetes mellitus | | 27 | 12.7 |
| Liver cirrhosis | | 15 | 7.0 |
| Salvage surgery * | No | 107 | 50.2 |
| | Yes | 102 | 47.9 |
| Preoperative radiotherapy | No | 126 | 59.2 |
| | Yes | 87 | 40.8 |
| Cancer location | Oropharynx | 8 | 3.8 |
| | Hypopharynx | 112 | 52.6 |
| | Larynx | 91 | 42.7 |
| | Thyroid | 2 | 0.9 |
| Neck dissection | No | 26 | 12.2 |
| | Ipsilateral | 102 | 47.8 |
| | Bilateral | 85 | 40.0 |
| Reason of surgery | Primary treatment | 107 | 50.2 |
| | Residual tumor | 56 | 26.3 |
| | Recurrent tumor | 46 | 21.6 |
| | Non-functional | 4 | 1.9 |
| Wound closure | Primary closure | 48 | 22.5 |
| | Free flap reconstruction | 165 | 77.5 |
| Design of free flap reconstruction | Patch on | 101 | 47.4 |
| | Tubing | 64 | 30.1 |
| Postoperative major wound infection | | | 57 |

[a]BMI: Body mass index

[b]NTD: New Taiwan dollars; 1 U.S. dollar = 30.2 NTD (according to the Bank of Taiwan, as of June 3, 2020)

* not include non-functional surgery

The average preoperative serum hemoglobin level was 12.901 g/dL (range: 7.7–17.8). The average postoperative serum hemoglobin level was 11.637 g/dL (range: 8.35–15.4). The average preoperative serum albumin level was 4.075 g/dL (range: 2.7–4.95). The average postoperative serum albumin level was 3.1 g/dL (range: 1.9–4.3). In this cohort, 27 patients (12.7%) had type II diabetes mellitus, and 15 patients (7.0%) had liver cirrhosis (14 patients had Child class A liver cirrhosis; the other patient had Child class B liver cirrhosis according to the Child-Pugh score).

The most common tumor subsite was the hypopharynx (n = 112, 52.6%), followed by the larynx (n = 91, 42.7%), base of tongue (n = 8, 3.8%), and thyroid (n = 2, 0.9%). All base of tongue cancers in this cohort were p16-negative tumors. Total laryngectomy which was

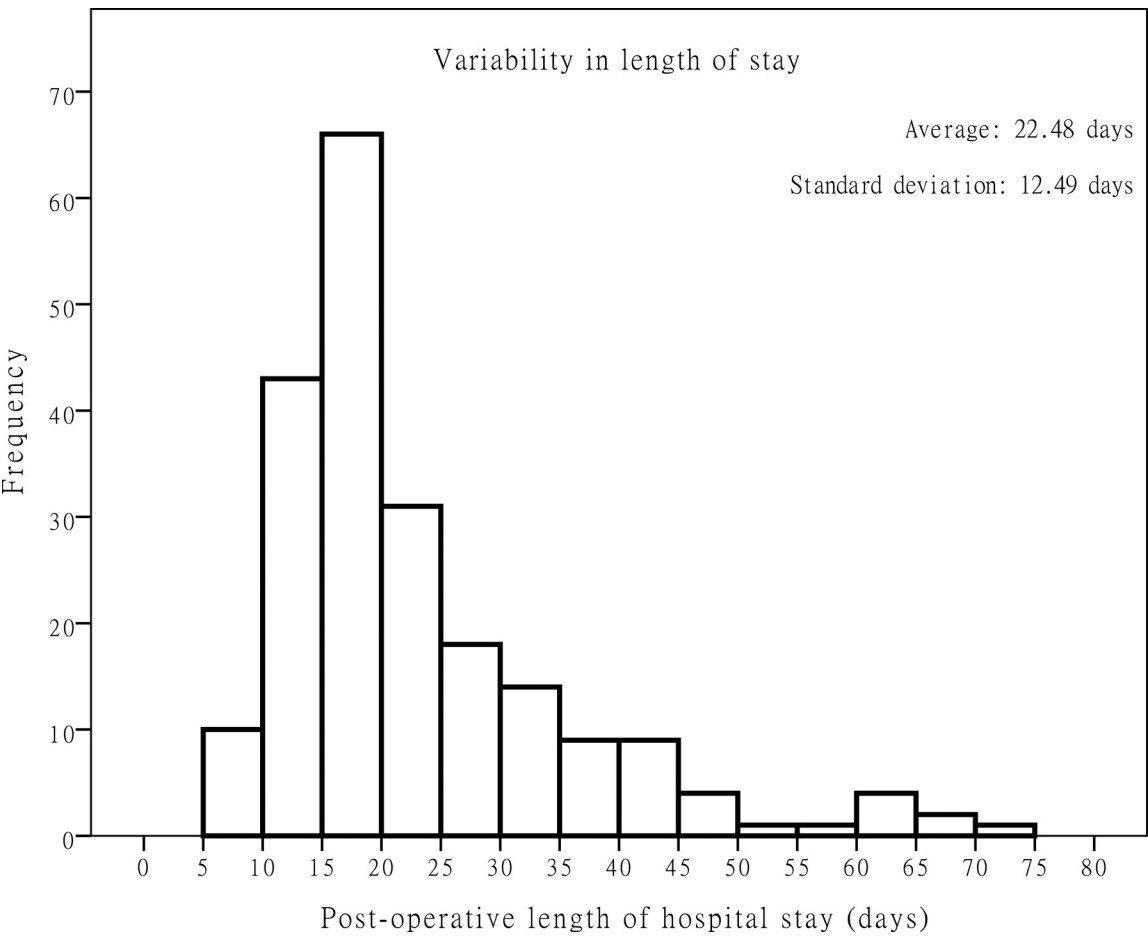

**Fig 1. Variability in length of hospital stay.**

performed among patients of base of tongue cancer or thyroid cancer was due to the direct involvement of larynx by tumor. The most histopathologic cancer type in this population was squamous cell carcinoma; the other 2 patients had papillary thyroid carcinoma.

There were 107 patients (50.2%) who underwent TL as the primary treatment, 102 patients (47.9%) who underwent salvage surgery for recurrent or persistent head and neck cancer after definite treatment, and 4 patients (1.9%) who underwent TL due to non-functional larynx secondary to previous organ preservation therapy. There were 87 patients (40.8%) who had prior RT history. The radiation technique for these patients was intensity-modulated radiation therapy (IMRT). The primary radiation dose was between 6,600 and 7,000 cGy (2 cGy/fraction).

In terms of operative method, ipsilateral neck dissection was performed in 102 patients (47.8%), 85 patients (40%) underwent bilateral neck dissections, and the other 26 patients (12.2%) didn't receive neck dissection because of negative nodal status after CCRT or RT. Among all patients, 165 (77.5%) underwent reconstruction with free flap transfer, including 160 anterolateral thigh and 5 anteromedial thigh flaps; 101 patients were reconstructed by the patch-on method; the other 64 patients were reconstructed by the flap-tubing method.

## Mortality and immediate surgical complications

Overall, 2 patients died in this hospitalization (2/213 = 0.9%). One patient had hypopharyngeal cancer (ypT4bN0M0), underwent salvage TL with free flap reconstruction for a persistent

tumor after CCRT, and died on postoperative day 47 due to pharyngocutaneous fistula (PCF) related carotid blowout. The other patient, who had liver cirrhosis history (Child-Pugh classification A), underwent salvage TL for persistent supraglottic cancer (ypT2N3bM0), restaged according to the eighth edition of the AJCC system) with free flap reconstruction after the failure of CCRT. This patient developed postoperative pneumonia and wound infection with PCF and died on postoperative day 58 due to severe sepsis. There were 5 patients (5/165 = 3%) who had free flap failure and in whom plastic surgeons redid another free tissue transfer. Nine patients (9/165 = 5.45%) had anastomosis site leakage with an acute bleeding episode and required surgical exploration.

**Major wound infection.** Major wound infection was defined as a wound that needed to be debrided and managed in the operating room. The incidence of postoperative major wound infection was 26.8% (57/213). Donor site wound infection or complication was not noted in this cohort.

## Length of hospital stay

Several factors influencing prolonged postoperative LOS were selected for univariate analysis (Table 2). Prior RT history ($p = 0.007$), free flap reconstruction for pharyngeal defect ($p < 0.001$), postoperative lower serum albumin level ($p = 0.025$), and postoperative anemia ($p = 0.04$) were all significantly associated with higher probability of prolonged LOS. Logistic regression analysis was then performed, using these significant factors in univariate analysis. In this model, postoperative serum albumin level was a significant independent predictor of prolonged LOS (OR 2.477, 95% CI 1.189–5.163, $p = 0.015$). In addition, prior RT history significantly increased the probability of postoperative LOS as compared to no history of RT (OR 2.194, 95% CI 1.228–3.917, $p = 0.08$) (Table 3).

## Inpatient medical cost

The association between clinical variables and inpatient medical cost is shown in Table 2. Free flap reconstruction for pharyngeal defect ($p < 0.001$), preoperative lower serum albumin level ($p = 0.04$), postoperative lower serum albumin level ($p < 0.001$), and history of liver cirrhosis ($p = 0.037$) were all significantly associated with higher probability of high inpatient medical cost. Logistic regression analysis was then performed, using these significant factors in univariate analysis. In this model, postoperative hypoalbuminemia was a significant independent predictor of higher inpatient medical cost (OR 6.671, 95% CI 1.927–23.093, $p = 0.003$). In addition, a wound that needed free flap reconstruction significantly increased the probability of high inpatient medical cost as compared to the group whose wounds were closed primarily (OR 5.011, 95% CI 1.657–15.156, $p = 0.04$) (Table 4).

Patients with postoperative hypoalbuminemia had significantly longer postoperative LOS than those patients with higher postoperative albumin levels ($\geqq$3.5 g/dL; Mann-Whitney test, $p = 0.001$). On average, patients with lower postoperative albumin levels ($<$3.5 g/dL) had 23.49 days of hospitalization after surgery, which is 35% more days compared to patients with higher postoperative albumin levels ($\geqq$3.5 g/dL), whose LOS was 17.39 days on average. As the length of hospitalization increased, the patient's total inpatient medical bill increased correspondingly. The total inpatient medical costs of patients with lower postoperative albumin levels ($<$3.5 g/dL) were significantly higher than those of patients with higher postoperative albumin levels ($\geqq$3.5 g/dL; Mann-Whitney test, $p < 0.001$). Patients with lower postoperative albumin levels ($<$3.5 g/dL) had an average medical cost of 404,875 NTD (about 13,406 U.S. dollars), which is 48% higher than that of patients with higher postoperative albumin levels ($\geqq$3.5 g/dL), who had an average cost of 272,847 NTD (about 9,035 U.S. dollars) (Table 5).

**Table 2. Univariate analysis of factors impacting postoperative hospital stay and total inpatient medical cost.**

| Variable | | Postoperative hospital stay(≧18 days) | p | Total inpatient medical cost (≧ 379,000 NTD[b]) | p |
|---|---|---|---|---|---|
| Cancer location | Oropharynx* | - | 0.521 | - | 0.807 |
| | Hypopharynx | 58 | | 40 | |
| | Larynx | 43 | | 31 | |
| | Others* | - | | - | |
| Salvage surgery | No | 46 | 0.062 | 35 | 0.327 |
| | Yes | 57 | | 40 | |
| Preoperative radiotherapy | No | 53 | 0.007 | 40 | 0.203 |
| | Yes | 53 | | 35 | |
| Neck dissection | No | 11 | 0.417 | 6 | 0.167 |
| | Yes | 95 | | 69 | |
| Reason of surgery | Primary treatment | 46 | 0.169 | 35 | 0.619 |
| | Residual tumor | 32 | | 22 | |
| | Recurrent tumor | 25 | | 18 | |
| | Non-functional* | - | | - | |
| Wound closure | Primary closure | 10 | <0.001 | 4 | <0.001 |
| | Flap reconstruction | 96 | | 71 | |
| Design of free flap reconstruction | Patch on | 55 | 0.223 | 38 | 0.078 |
| | Tubing | 41 | | 33 | |
| Preoperative albumin level (g/dL) | < 3.5 | 9 | 0.144 | 8 | 0.04 |
| | ≧ 3.5 | 81 | | 56 | |
| Postoperative albumin level (g/dL) | < 3.5 | 88 | 0.025 | 70 | <0.001 |
| | ≧ 3.5 | 14 | | 3 | |
| BMI[a] (Kg/m2) | < 23 | 73 | 0.303 | 54 | 0.069 |
| | ≧ 23 | 31 | | 18 | |
| Liver cirrhosis | No | 95 | 0.058 | 66 | 0.037 |
| | Yes | 11 | | 9 | |
| Diabetes mellitus | No | 93 | 0.857 | 66 | 0.827 |
| | Yes | 13 | | 9 | |
| Preoperative hemoglobin level (g/dL) | < 13 | 53 | 0.946 | 43 | 0.127 |
| | ≧ 13 | 53 | | 32 | |
| Postoperative hemoglobin level (g/dL) | < 13 | 96 | 0.04 | 69 | 0.05 |
| | ≧ 13 | 10 | | 6 | |

* subgroup which not include for analysis

[a]BMI: body mass index

[b]NTD: New Taiwan dollars; 1 U.S. dollar = 30.2 NTD (according to the Bank of Taiwan, as of June 3, 2020)

**Table 3. Multiple regression analysis of factors impacting longer postoperative hospitalization (≧ 18 days) of all patients.**

| Factor | | Odds ratio | 95% Confident Interval | | p value |
|---|---|---|---|---|---|
| Postoperative Albumin (g/dL) | ≧ 3.5 | 1 | | | .015 |
| | < 3.5 | 2.477 | 1.189 | 5.163 | |
| Preoperative radiotherapy | No | 1 | | | .008 |
| | Yes | 2.194 | 1.228 | 3.917 | |

**Table 4. Multiple regression analysis of factors impacting higher total inpatient medical cost (≧ 379,000 NTDa).**

| Factor | | Odds ratio | 95% Confident Interval | | p value |
|---|---|---|---|---|---|
| Postoperative Albumin (g/dL) | ≧ 3.5 | 1 | | | .003 |
| | < 3.5 | 6.671 | 1.927 | 23.093 | |
| Wound closure | Primary closure | 1 | | | .004 |
| | flap reconstruction | 5.011 | 1.657 | 15.156 | |

[a] NTD: New Taiwan dollars; 1 U.S. dollar = 30.2 NTD (according to the Bank of Taiwan, as of June 3, 2020)

## Discussion

In the previous studies, BuSaba and Schaumberg described multiple factors, such as several comorbid conditions, longer operative time, intraoperative blood transfusion and postoperative complications, were significantly associated with increased length of stay in elective major head and neck surgeries [6]. In different laryngeal preservation treatments in glottic Cancer, Mandelbaum et al. had also found that open surgery, endoscopic surgery was associated with reduced hospital charges than primary chemoradiation therapy [7]. Dombrée et al had studied the surgical charge by different surgical methods for total laryngectomy, which demonstrated the surgical cost was more expensive in TORS (the average surgical cost was 6,767 Euro) than open approach (the average surgical cost was 3,581 Euro) [8].

In our cohort, almost 40% of patients had had history of prior radiation therapy. Major wound infection occurred in 26.8% of patients. We found that length of stay was highly associated with postoperative hypoalbuminemia and history of prior radiation therapy. Lastly, high inpatient medical cost was significantly associated with postoperative hypoalbuminemia and wound closure by free flap for neopharyngeal defect.

The lengths of postoperative hospital stay could be various in different countries due to the distinct type of funding in each country and the various post-hospitalization structures [9]. Most patients discharged from the hospital usually going back home in Taiwan directly.

Goepfert et al. demonstrated that preoperative RT significantly influenced the length of hospital stay [2]. Tissue blood flow dysfunction may occur after radiotherapy and lead to poor wound healing status [10]. Among patients who diagnosed with HNSCC and need surgical salvage after definite radiotherapy, as Sassler et al. reported in their study, the incidence of wound complications was up to 64% [11]. In the RTOG Trial 91–11, PCF rates after salvage total laryngectomy ranged from 15 to 30% [12]. Other studies showed that PCF rates were

**Table 5. Comparison of central tendency between different postoperative albumin level among postoperative length of hospital stay and total inpatient medical cost.**

| | Postoperative length of hospital stay (days) | | Total inpatient mediation cost (NTD [a]) | |
|---|---|---|---|---|
| | Postoperative albumin< 3.5 g/dL | Postoperative albumin≧3.5 g/dL | Postoperative albumin< 3.5 g/dL | Postoperative albumin≧3.5 g/dL |
| Number | 164 | 41 | 164 | 41 |
| Range | [8, 69] | [7, 40] | [136725, 1192203] | [121275, 454668] |
| Mean | 23.49 | 17.39 | 404875 | 272847 |
| Mean rank | 110.00 | 75.00 | 114.28 | 57.87 |
| Mann-Whitney U statistic | 2214.0 | | 1511.5 | |
| p value | 0.001 | | <0.001 | |

[a] NTD: New Taiwan dollars; 1 U.S. dollar = 30.2 NTD (according to the Bank of Taiwan, as of June 3, 2020)

relatively high, range from 15 to 50% for salvage surgery after definite radiotherapy or chemoradiotherapy [5, 11, 12]. Our cohort also revealed similar results: that history of radiotherapy increased the wound infection rate and prolonged hospital stays.

Another possible factor increased the rate of wound infection and PCF formation may be low thyroid function status perioperatively, especially in patients had prior RT history. In previous study, Rosko et al. described postoperative hypothyroidism independently predicts postoperative wound-healing complications including PCF formation in patients who underwent salvage total laryngectomy after RT/CRT [13]. In our cohort, only 41 patients (41/213 = 19.2%) had checked thyroid function status in postoperative period. Fifteen patients were in clinical hypothyroidism, six patients were in subclinical hypothyroidism and other twenty patients were in euthyroid status. All patients diagnosed as hypothyroidism after surgery would be immediately treated with thyroxine according to our guideline. The subgroup analysis didn't show significance in the association between thyroid function status and clinical outcomes, including postoperative hospital stay and inpatient medical cost.

The management of pharyngeal mucosa defect after TL depends on the width of residual pharyngeal wall mucosa. If it is not adequate for primary closure or the circulation over the pharyngeal mucosa is poor after RT/CRT, reconstruction with local flap or free flap is inevitable. In turn, this prompts a lengthier hospital stay, higher medical expenses, and more potential morbidities, according to the data from this cohort.

Malnutrition is a well-known factor that related to poor wound healing. Hypoalbuminemia, as one of marker of malnutrition, also indicates prolonged hospital stay [14, 15]. In the present study, however, postoperative albumin level had a stronger association with prolonged hospital stay than any other nutrition-related marker. In the Takahara's study, postoperative hypoalbuminemia has a stronger association with longer hospital stay than other nutrition-related marker [16]. Our previous studies showed similar results: that postoperative hypoalbuminemia is a useful indicator of the development of postoperative complications and prolonged postoperative hospital stay among patients who underwent tumor excision and free flap reconstruction for an advanced stage of HNSCC [17].

A locally advanced tumor stage over the larynx or hypopharynx may cause poor nutrition status before treatment. In addition, previous RT/CRT for locally advanced tumors may further decrease the nutrition reserve after this treatment cycle. This study demonstrated that lower albumin status before surgical intervention would increase the medical cost, complications after operation, and the hospital stay.

There are several limitations that should be addressed in the current study. First, this is a retrospective, single-institute study and selection bias in patient and data collection may happen. Second, due to the different indication of surgery, cancer location, histological type of cancer, and prior radiotherapy or not, the heterogeneity between patients was still high. Third, another important weak point was lack of thyroid function status in most our patients in our cohort.

## Conclusions

Previous RT history and postoperative lower albumin level show impacts on the length of hospital stay. In addition, the application of free flap reconstruction for pharyngeal defect and postoperative lower albumin status increase the inpatient medical cost among patients who undergo total laryngectomy for laryngeal cancer/hypopharyngeal cancer. In our cohort, the inpatient medical cost was 48% higher and length of stay after surgery was 35% longer among hypoalbuminemia patients.

## Supporting information

**S1 Study dataset. Analytical dataset used in the study.** https://doi.org/10.6084/m9.figshare.12568565.v1.
(XLSX)

## Author Contributions

**Conceptualization:** Ming-Hsien Tsai, Hui-Ching Chuang, Chih-Yen Chien.

**Data curation:** Ming-Hsien Tsai, Fu-Min Fang, Tai-Lin Huang, Tai-Jan Chiu, Shau-Hsuan Li.

**Formal analysis:** Yu-Tsai Lin, Hui Lu.

**Methodology:** Ming-Hsien Tsai, Hui-Ching Chuang, Yu-Tsai Lin, Tai-Jan Chiu, Chih-Yen Chien.

**Resources:** Yu-Tsai Lin, Fu-Min Fang.

**Software:** Hui Lu.

**Supervision:** Tai-Lin Huang, Chih-Yen Chien.

**Validation:** Chih-Yen Chien.

**Visualization:** Fu-Min Fang.

**Writing – original draft:** Ming-Hsien Tsai.

**Writing – review & editing:** Ming-Hsien Tsai, Hui-Ching Chuang, Chih-Yen Chien.

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
