## [Decision Letter · Decision Letter 0]

17 Jun 2020

PONE-D-20-17243

Predictors of hospital expenses and hospital stay among patients undergoing total laryngectomy: Cost effectiveness analysis

PLOS ONE

Dear Dr. Chien,

Thank you for submitting your manuscript to PLOS ONE. After careful consideration, we feel that it has merit but does not fully meet PLOS ONE’s publication criteria as it currently stands. Therefore, we invite you to submit a revised version of the manuscript that addresses the points raised during the review process.

We look forward to receiving your revised manuscript.

Kind regards,

Giovanni Cammaroto

Academic Editor

PLOS ONE

Journal Requirements:

2. In the ethics statement in the manuscript and in the online submission form, please provide additional information about the patient records used in your retrospective study. Specifically, please ensure that you have discussed whether all data/tissue samples  were fully anonymized before you accessed them and/or whether the IRB or ethics committee waived the requirement for informed consent. If patients provided informed written consent to have data from their medical records used in research, please include this information.

3. We noticed minor instances of text overlap with the following previous publication(s), which need to be addressed:

(1) https://journals.sagepub.com/doi/10.1177/0194599810393117

(2) https://www.sciencedirect.com/science/article/abs/pii/S1368837509009877?via%3Dihub

(3) https://www.tandfonline.com/doi/abs/10.1080/00016489.2018.1532108?journalCode=ioto20

(4) https://www.jstage.jst.go.jp/article/circj/82/10/82_CJ-18-0289/_article

The text that needs to be addressed involves the Discussion section (lines 219-224, 235-237, 240-248, 255-265).

In your revision please ensure you cite all your sources (including your own works), and quote or rephrase any duplicated text outside the methods section. Further consideration is dependent on these concerns being addressed.

Reviewers' comments:

Reviewer's Responses to Questions

**Comments to the Author**

1. Is the manuscript technically sound, and do the data support the conclusions?

Reviewer #1: Yes

Reviewer #2: Yes

2. Has the statistical analysis been performed appropriately and rigorously? 

Reviewer #1: Yes

Reviewer #2: Yes

3. Have the authors made all data underlying the findings in their manuscript fully available?

Reviewer #1: No

Reviewer #2: Yes

4. Is the manuscript presented in an intelligible fashion and written in standard English?

Reviewer #1: Yes

Reviewer #2: Yes

5. Review Comments to the Author

Reviewer #1: A well-written article.

Results are very difficult to extrapolate, because as the authors highlight cost related to surgery varies according to each country.

However, add interesting information in order to increase savings related to healthcare.

Reviewer #2: please see attachment

6. PLOS authors have the option to publish the peer review history of their article (what does this mean?). If published, this will include your full peer review and any attached files.

Reviewer #1: Yes: Carlos M Chiesa-Estomba

Reviewer #2: Yes: Jerome R. Lechien

---

## [Author Response · Author response to Decision Letter 0]

26 Jun 2020

We sincerely appreciate the constructive comments by your esteemed reviewers and we believe it has enhanced the clarity of our work. We have made the changes accordingly, along with the point-by-point response below. Once again we thank the editorial board of PLOS ONE and we hope to see our work published in this prestigious journal in the near future. 

Many thanks

Reply to reviewers’ comments

To Reviewer #1:

A well-written article.

Results are very difficult to extrapolate, because as the authors highlight cost related to surgery varies according to each country.

However, add interesting information in order to increase savings related to healthcare.

Response: 

We thank the reviewer for this great point. It is true that different health care systems in different countries will affect the length of postoperative hospital stay and inpatient medical expenses after total laryngectomy. We add this in the discussion. (line 241-242)

 

To Reviewer #2:

The authors investigated the predictive factors of PO hospital stay & medical cost in patients who benefited from TL. They identified several factors associated with the PO hospital stay or medical cost, including postoperative Hb level, albumin, preoperative RT (salvage TL), cirrhosis history or wound closure method. The study is well-conducted and designed and helpful for OTOHNS surgeons. The paper is short and to-the-point, which is particularly important for a potential publication in the journal. 

Some point has however been addressed to improve the manuscript. 

Abstract:

1. Develop the method section: state the studied outcomes (preop RT, etc.). 

Response: We thank the reviewer for this suggestion. We added this description about studied outcomes in section of abstract. (line 7 - 12)

Introduction:

2. To the point, well organized. 

Response: We thank the reviewer for your appreciation.

Methods:

3. What were your indication to make a flap and which flap were used ? Specify the exclusion criteria. 

Response: Free flap transfer for pharyngeal defect reconstruction after total laryngectomy would be performed if pharyngeal defect couldn’t primarily close without tension. The flap was most often harvested from the anterolateral thigh (n=160) followed by the anteromedial thigh (n=5) in our cohort. Patients who received partial laryngectomy or partial laryngopharyngectomy or patient didn’t receive total laryngectomy surgical procedure in our hospital would be excluded from our study. We have also updated the manuscript in the section of Method. (line 67 - 71 and line 153 - 156). 

4. Line 80: you already define BMI in the abbreviation (line 70), just keep in this line BMI. 

Response: We thank the reviewer for this insightful observation. We had change word from “Body mass index (BMI, kg/m2)” to “BMI”. (line 85)

Results:

5. line 114 = the cost is presented in Taiwan money and U.S. dollar. Please, present in the brackets the Euro value for European Physicians. Thus, you cover both U.S. and Europe. 

Response: Thank you very much for this useful recommendation. We had added the information of Taiwan Dollar to Euro Exchange Rate in our revised manuscript. (line 122 - 123)

6. Briefly specify the RT protocol (number of Gy, fraction, etc.)

Response: We thank the reviewer for this suggestion. The radiation technique for these patients was intensity-modulated radiation therapy (IMRT). The primary radiation dose was between 6,600 and 7,000 cGy (2 cGy/fraction). We had added this description in the section of Result. (line 149 - 150)

Discussion:

7. Another biological point that is important to consider is the thyroid status. In practice, we usually observed that hypothyroidism patients have wound healing disorder and longer hospital stay. The lack of consideration of this point is a limitation that has to be addressed. 

Response: We thank the reviewer for this great point. In previous study, Rosko et al. described postoperative hypothyroidism independently predicts postoperative wound-healing complications including pharyngocutaneous fistula formation in patients who underwent salvage total larygectomy [1]. In our cohort, most patients lack of thyroid function status in laboratory exam before/after total laryngectomy. Only 10 patients (10/213=4.7%) had thyroid function status before total laryngectomy and all of them were in euthyroid status. In postoperative period, 41 patients (41/213=19.2%) had checked thyroid function status during the hospitalization. Of them, 20 patients were in euthyroid status, 15 patients were in clinical hypothyroidism (decreased serum free T4 level and elevated serum TSH level) and 6 patients were in subclinical hypothyroidism (elevated serum TSH level and normal serum free T4 level). All patients diagnosed with hypothyroidism had been immediately treated with thyroxine according to our guideline. We have performed additional subgroup analyses between different postoperative thyroid function status. The association between thyroid function status and our interested outcome, including postoperative hospital stay and inpatient medical cost was showed in Table A. The univariate analysis didn’t show significance. The reason may be (1) the small number (n=41) in this analysis and (2) temporary low thyroid functional level postoperatively, because thyroxine supplement had been prescribed after hypothyroidism diagnosed. We have also updated the manuscript to reflect these findings. (line 253 – 263, 286 – 287)

8. Develop the limitation paragraph, heterogeneity between patients (indication/tumor localization, RT) etc. that may consist of heterogeneity in the study. The inclusion of non-SCC (papillary thyroid) is another bias. 

Response: We thank the reviewer for this suggestion. We had added the limitation paragraph about this study in section of discussion (line 283 – 287)

Conclusion:

9. Remove “in total”. The sentence could be: Previous RT history and postoperative lower albumin level significantly impact the length of hospital stay.

Response: We thank the reviewer for this suggestion. We had changed this sentence in section of conclusion to “Previous RT history and postoperative lower albumin level significantly impact the length of hospital stay.” (line 290)

Reference:

1. Rosko AJ, Birkeland AC, Bellile Eet al. Hypothyroidism and Wound Healing After Salvage Laryngectomy. Ann Surg Oncol 2018; 25:1288-1295.

---

## [Editor Report · Decision Letter 1]

30 Jun 2020

Predictors of hospital expenses and hospital stay among patients undergoing total laryngectomy: Cost effectiveness analysis

PONE-D-20-17243R1

Dear Dr. Chien,

We’re pleased to inform you that your manuscript has been judged scientifically suitable for publication and will be formally accepted for publication once it meets all outstanding technical requirements.

Kind regards,

Giovanni Cammaroto

Academic Editor

PLOS ONE
---

## [Editor Report · Acceptance letter]

6 Jul 2020

PONE-D-20-17243R1 

Predictors of hospital expenses and hospital stay among patients undergoing total laryngectomy: Cost effectiveness analysis 

Dear Dr. Chien:

I'm pleased to inform you that your manuscript has been deemed suitable for publication in PLOS ONE. Congratulations! Your manuscript is now with our production department. 

Kind regards, 

on behalf of

Dr. Giovanni Cammaroto 

Academic Editor

PLOS ONE